# A Blended Learning Approach for an Introductory Computer Science Course

**Anna Förster** [1,*] **, Jens Dede** [1] **, Asanga Udugama** [1] **, Alexander Förster** [2] **, Daniel Helms** [1] **, Louis Kniefs** [1] **, Julia Müller** [1] **, Lars Gerken** [1] **, Franziska Richter** [3] **and Jan Kulmann** [3]

1 Sustainable Communication Networks, University of Bremen, 28359 Bremen, Germany; jd@comnets.uni-bremen.de (J.D.); adu@comnets.uni-bremen.de (A.U.); dhelms@uni-bremen.de (D.H.); kniefs@uni-bremen.de (L.K.); julia.mueller@uni-bremen.de (J.M.); lagerken@uni-bremen.de (L.G.)
2 Roboter und Technik, 28213 Bremen, Germany; axf@roboter-und-technik.de
3 Zentrum für Multimedia in der Lehre, University of Bremen, 28359 Bremen, Germany; franziska.richter@vw.uni-bremen.de (F.R.); kulmann@zmml.uni-bremen.de (J.K.)
* Correspondence: anna.foerster@uni-bremen.de

**Abstract:** In this paper, we present our experience with redesigning an introductory computer science course for (electrical) engineers with blended learning concepts. It is a large mandatory course with eight credit points over the course of two semesters, where first year electrical engineers learn how to program with Arduino, C and Python. Additionally, they need to cover basic computer science concepts such as binary numbers, Boolean algebra, encodings, finite state machines and object-oriented programming. We designed the course to cover the fields that are most relevant to our students' future careers. With the new format, students declare they are much more engaged with the course, they drop the course less often and they actually achieve better exam results. We completely re-structured the course, introduced inverted classroom elements and hackathons and we continuously optimized and adapted the course. The main focus is on hands-on-experience and teamwork, which we mostly achieve by the use of hackathons. In this paper, we described the contents and teaching concepts of the course and we discussed the achieved results.

**Keywords:** blended learning; inverted classroom; programming; Arduino; teaching; electrical engineering; Python; object-oriented programming; data analysis

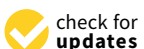



## 1. Introduction

In Germany, traditional teaching with lectures and tutorials has been the predominant method for centuries. The word for "lecture" in German is "Vorlesung", which literally means "reading to somebody" and still describes how lectures are also held nowadays. On the opposite, the inverted classroom [1] enables students to acquire knowledge before classes and to focus on practical exercises and more complex connections during class. Inverted classroom flips the learning process: students first independently acquire knowledge and then discuss and practice this knowledge with the class instructor. This approach has already been applied before the online learning era, i.e., by giving mandatory reading assignments before class.

Blended learning [2] is a similar concept with a slightly different focus. It combines the best from online work and in-class work, i.e., the normal learning process is blended with videos, online assignments, forum discussions, and other online resources. To some extent, nowadays, both concepts significantly overlap, as most of the flipped learning tasks are also mostly online activities.

This paper presents our experience with redesigning an introductory computer science course for electrical engineers with blended learning concepts. The course we are describing here is part of an electrical engineering (EE) bachelor program in Germany. It takes place during the second and third semesters of the bachelor program, with six and three ECTS

(European Credit Transfer and Accumulation System (https://ec.europa.eu/education/resources-and-tools/european-credit-transfer-and-accumulation-system-ects_en, accessed on 15 July 2021)), which corresponds to approximately 4 and 2 h per week. Before the redesign, it used to be taught as a lecture-based class, where content is introduced during lectures, with extra tutorials including in-class assignments and homework assignments. The general experience was not satisfactory: students would not listen carefully to the lecture and would come not prepared for the tutorials. We especially had problems with a high drop-out rate and low scores and low interest in the subject in general.Please confirm whether these 2 emails belong to the correspondence author.

In this paper, we present a thorough analysis of the challenges we encountered, the envisioned learning goals, and defined research questions to identify the suitability of blended learning for our goals. Our results show that blended learning is undoubtedly an up-and-coming alternative to traditional lecturing, with drop-out rates decreasing and final scores slightly increasing. However, we have also met a range of different new problems and challenges, which we also discuss in this paper and which can be of great value to other instructors.

Subsequently, we first discuss some related works and explain the background of our work in Section 2. Then, we define our problem, our challenges and our methodology in Section 3. In Section 4, we explain the structure of the new blended learning course and its characteristic elements. Section 5 discusses some changes we implemented on the way while receiving feedback from students and their achievements. Section 6 presents an evaluation and comparison before and after we implemented the blended learning concept. Section 7 discusses some remaining issues and how the concepts can be transferred to other courses and fields, while Section 8 concludes the paper.

## 2. Background

This section discusses some related works and background information necessary to fully understand our course's design and decisions.

### 2.1. Related Works

Electrical engineering (EE) is a traditional study program in Germany (Elektrotechnik in German). A search for available study programs in EE at the popular German university information website HOCHSCHULKOMPASS.DE provides over 500 options. Almost all of them also include some portion of Computer Science, mostly programming courses. The exact extent varies between the different universities. For example, the University of Bremen has only nine ECTS CS courses ( www.uni-bremen.de/fb1/studium, accessed on 15 July 2021), while RWTH Aachen has 24 ECTS of CS-focused subjects (www.elektrotechnik.rwth-aachen.de, accessed on 15 July 2021). The module descriptions are typically available online (in German) for all EE programs, but the teaching methods are usually not specified. To the best of our knowledge, no experience reports specifically exist for the usage of inverted classroom or blended learning in EE in Germany.

Currently, the flipped classroom concept is a popular teaching method in Germany in many universities and topics ([3] and references within). One prominent proponent is Prof. Hardtke, who also maintains a popular blog about it [4]. A prominent international example of the flipped classroom is Khan's academy for mathematics [5].

A range of articles also suggest that the inverted classroom is an effective teaching method for computer science [6,7] and for engineering [8]. Very similar problems to ours (discussed in detail in Section 3) were identified and tackled with blended learning in the work of [9], which describes a use case in a statistics course for CS students. The results were also similar to ours: interest and motivation rise, drop rates decrease.

In our course, we focus on two programming languages: Arduino/C and Python. These are recommended by the IEEE for EE students [10] and are popular choices worldwide. The Arduino platform has also been thoroughly evaluated for teaching purposes for various engineering [11,12] and programming courses [13,14], with great success. We

introduced the usage of Arduino well before the redesign of the course described here, also with great success. Arduino builds a natural bridge between EE and CS and students perceive it as very valuable, not only for their studies but also for hobbies (see more about the students' evaluations in Section 6).

### 2.2. Teaching Environment

The teaching environment in German universities is quite different from those in other countries, especially in the US. Without a loss of generality, we discuss here how teaching is organized in our faculty.

The study programs we target here are the Bachelor of Electrical Engineering and the Bachelor of Electrical Engineering with Management. The studies are organized into modules, typically consisting of a few courses. Our module is called Foundations of Computer Science and consists of two courses or parts, with six and three ECTS, respectively. Each module is assigned with a responsible professor or a senior researcher, who is usually also teaching the courses. Each working group (in German: Lehrstuhl or chair) has a few state-financed researchers. The exact number mainly depends on the research field (engineering disciplines tend to have more, social sciences less), the professor's experience. Professors and researchers financed by the state must teach a particular number of hours per week. In Bremen, it is typically 9 h for professors and 4 h for researchers (one teaching hour per week corresponds to 14 h per semester or 28 h per year—these do not include preparation, exams, or assignment grading, but only pure lecturing time.). Furthermore, most faculties have a budget for tutors. The number of tutors per course is calculated depending on the number of students. Usually, 25–30 students are assigned per tutor. This tutor leads exercises and practical sessions, corrects assignments, and answers questions.

This system is quite different from other countries, where tutors are rare and lecturers need to optimize their time significantly. This deficit has led to different solutions in teaching, including chatbots, automatic grading and other computerized solutions. While such solutions are also applicable in Germany, personal supervision offers a more personalized approach [15], where individual problems can be addressed faster and more efficiently. This also offers a more flexible way of learning, where new exercises and topics can be introduced faster.

## 3. Problem Statement, Goals and Methodology

In this section, we first describe the challenges we encountered during two years of teaching the class. Then, we describe our identified goals and finally define our methodology, research questions and metrics.

### 3.1. Challenges

We have been teaching the course several years before the transition and faced various challenges:

- Very disparate technical backgrounds among students: High school programs in Germany still do not include mandatory programming or computer science courses. Thus, some first-year students start with a solid technical background coming from elective courses or extra-curricular activities, in contrast to others who do not have any programming or computer science experience—to the extent of having never used a computer before (however, they do use a smartphone.). This variety makes it very difficult to meet all students' learning requirements and match their levels.
- Very diverse cultural and language backgrounds: While most of our students come from German high schools and are proficient in German, around 10% come from abroad and have only mid-level German proficiency with almost no language experience. Cultural differences make it even harder for them to ask questions or mix with other students, hindering their learning process.
- The described course is the only CS course that students have in their EE studies, and contents have to be well aligned with their current and future requirements as

electrical engineers. With only eight ECTS in total, it is a challenging task to introduce basic CS concepts, select adequate programming language(s) and offer enough hands-on experience to solve CS problems on their own.

- The current format was not well suited for our needs, and all attempts to adapt it failed. Introducing programming principles and code snippets in lectures can be tedious and contra-intuitive, as students cannot directly practice their new skills. Introducing short in-class assignments did help a little but could not cater to the needs of the weaker students, who needed more time.
- Students' everyday lives are currently changing dramatically, at least in Germany. Most of them have additional responsibilities in addition to their studies, such as small children at home, part-time jobs, or home care to relatives. Furthermore, most of them continue living with their families and typically live far away from the university, up to 150 km. All these hinder the students from being regularly present at the university. Combined with sub-optimally designed lectures, students prefer staying home and learning from secondary materials and not spending their time traveling and attending boring lectures.
- An excellent indicator of students' interest in the course and its topics is how many students come back later to our group to conduct their bachelor projects. In general, and especially true for German universities, introductory courses like the CS class discussed herein include some topics from the research or expertise areas of the research group teaching them. In our case, our main research focus lies on the Internet of Things. Thus, we always included examples and assignments in this area, sometimes only as a motivational scenario. However, students still did not come back for bachelor projects, which was a sign that we were not able to mobilize their interest.

### 3.2. Goals

First, we identified our goals following the revised Blooms' taxonomy of teaching [16], which defines six categories in the cognitive domain: remember, understand, apply, analyze, evaluate and create. For this introductory course, we decided to reach at least the category **remember** and to have elements of the category **apply**.

Following the recommendations of IEEE [10], we identified two languages for our students: Arduino/C and Python. This combination has several advantages. Arduino with C teaches the students how to handle real hardware and can be of great help and use for their further studies and careers. Arduino also bridges the gap between electrical engineering and computer science. Some of the students did not recognize this vital connection beforehand. Python enables us to teach the students the basics of object-oriented programming. Furthermore, it is the basis for many popular libraries, such as NumPy, matplotlib, TensorFlow. Last but not least, by teaching two programming languages, students acquire abstraction abilities and can better learn other programming languages on their own [17].

After gaining some experience with our students, their needs, and their skills, we identified several learning goals. These goals cater to their special needs as electrical engineers:

1. Students should use basic programming constructs such as variables, loops, functions, input/output, random numbers, user interaction;
2. Students should understand the basic CS concepts, such as the binary and hexadecimal systems, internal variable representation and conversion, overflow and underflow;
3. Students should understand the principle of including libraries and use external libraries in their code;
4. Students should understand the principle of object-oriented programming and should use and further develop existing classes. Note that the goal is not for them to master all aspects of OOP but only to use class-based libraries in their code;
5. Students should be able to process, analyze and visualize data sets, such as sensory data;

6. Students should understand and use several important algorithm design principles, such as final state machines and flow charts;
7. Students should learn how to use programming resources and documentation online and how to research tasks they do not know yet. Given that we only have a very limited time to teach CS principles and programming, it is essential to acquire skills on how to learn on their own later.

Furthermore, our most important organizational goal was to find a way to design efficient presence working time at the university while offering more time and space flexibility to the students. Similarly, our goal was to offer more individualized learning, catering to the needs of language barriers and different technical backgrounds.

### 3.3. Research Questions and Methodology

Our own existing experience with a hands-on course on programming [18–20] motivated us to change the format of the course completely.

We decided on a use case as a methodology, where we intended to try out a more practical and less time-fixed approach. Our methodology consisted of two parts: blended learning to reach the **understand** category and hackathons to achieve the **apply** category. At the same time, blended learning adapts well to problems like time flexibility, cultural and language background, speed of learning and current knowledge level. Each student can go at their own pace, repeat videos, and assignments, and look for more information online. A hackathon offers a relatively stressful environment where the students need to apply new knowledge to a given application in a limited amount of time. This situation motivates students to conduct the blended learning part and gives them the experience of group work and clearly defined success.

Thus, our research questions are:

- RQ1: Can a combination of blended learning and a hackathon improve the students' learning outcome? Measure: total grades per semester;
- RG2: Can a combination of blended learning and a hackathon decrease the number of drop-outs? Measure: number of drop-outs per semester;
- RG3: Can a combination of blended learning and a hackathon increase the number of bachelor projects conducted with our group later? Measure: number of bachelor projects per year.

We focused on three metrics: total grades achieved, percentage of students dropping the class, and the number of bachelor projects we supervised. We were able to measure the first two metrics immediately after each semester. The third one was delayed due to the timing of the bachelor projects and due to the COVID-19 pandemic. We will discuss the achieved results in Section 6.

## 4. Course Description

When re-designing our course, we decided to use a combination of teaching methodologies, but blended learning [21] is probably the one that describes our approach best. However, the approach is also based on the inverted classroom method [1].

The course is divided into six equally organized modules, two weeks each. Each module consists of online learning material for self-study, including videos and online exercises. At the end of the module, the students come for a 4 h-long hackathon, where they work on a more extensive programming exercise in groups. Each group is assigned to a tutor (3–4 groups per tutor), who is helping, discussing and taking notes about their progress for later evaluation. Tutors are assigned to different groups at every hackathon. The students do not know the hackathon assignment in advance, and need to submit their solutions at the end of the hackathon.

Each of the first five modules introduces new content. The last hackathon is a bonus one, with does not introduce new content but exhibits a broader structure, covering the complete semester.

The students are divided into two groups of 50 students each, coming in alternating weeks for organizational reasons.

Figures 1 and 2 summarize the organization and the topics covered for both semesters. Sample hackathon assignments are also provided.

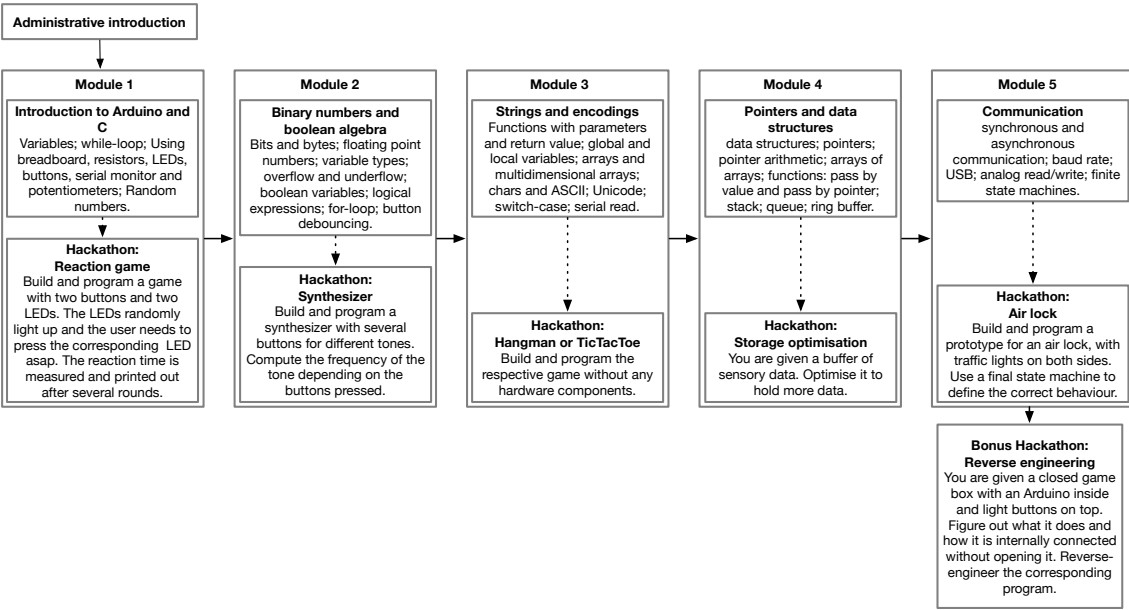

**Figure 1.** The organization of the first semester with content and sample hackathon assignments. The focus is on Arduino, C and basic programming concepts.

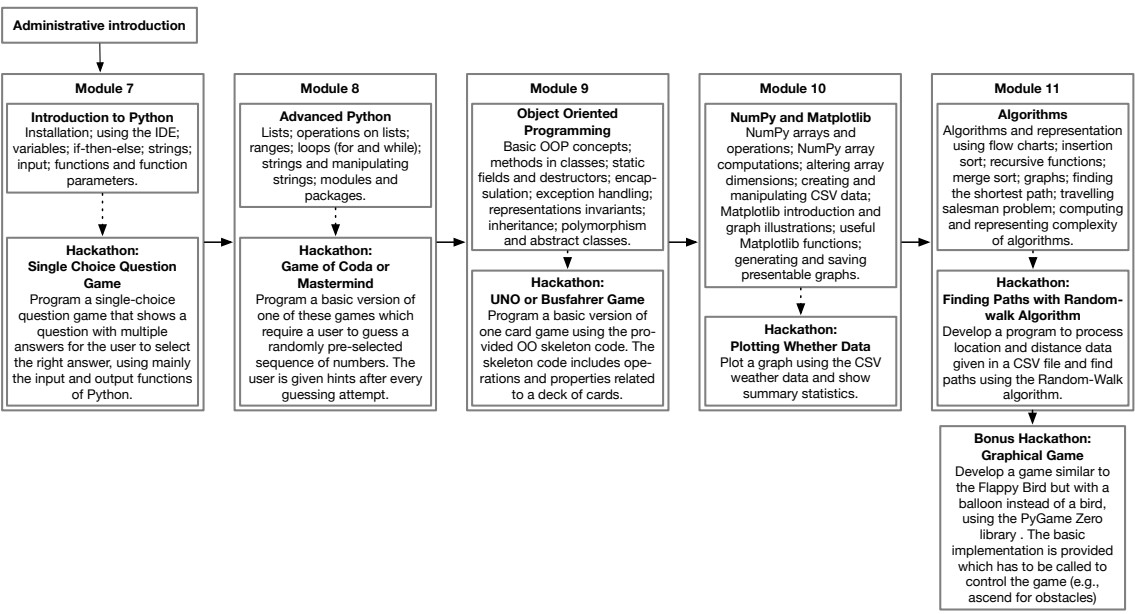

**Figure 2.** The organization of the second semester with content and sample hackathon assignments. The focus is on Python, OOP and data processing and visualization.

As one can see from the figures, the topics in each module are mixed general CS and programming concepts. In the first semester, the focus is on Arduino and hardware basics. This is needed in order to enable the students to start using Arduino components as soon as possible.

Each module consists of self-learning materials. For example, module 1 (Arduino and C basics) consists of 16 videos with lengths between 3 and 16 min, four reading assignments,

31 non-graded exercises, and nine graded exercises. Usually, 1–2 videos are followed by several exercises. The other modules have a similar structure and content amount.

The hackathon assignments were designed in a way that a student who has completed all self-study exercises, in fact, only needs to combine their solutions into a single project. For example, in the first module, we teach them variables, random numbers, buttons and LEDs and how to use the breadboard and the serial monitor. The hackathon assignment is to design a reaction game with two LEDs and two buttons. The LEDs light up randomly, and the user needs to press the corresponding button as soon as possible. The reaction time needs to be measured and printed out to the serial monitor after ten runs. Only the time measurement is left for research during the hackathon. Self-research is an essential aspect of our teaching concept and needs to be regularly practiced.

The grading of the course is a combination of the grades of the graded home assignments (10%), hackathons (40%) and a final multiple-choice exam at the end of each semester (50%). We deliberately decided to leave the exam as it is in order to be able to evaluate the effectiveness of the blended learning approach.

## 5. Continuous Changes through Experience

We implemented the new course for the first time during the winter semester of 2018/2019. Since then, we continuously optimized it through our own experience and observations but also through student evaluations and feedback. In this section, we describe some of the most important experiences and their solutions.

### 5.1. Graded Self-Study Exercises

In the first edition of the course, we observed that students gradually stopped preparing for the hackathons. We gathered some feedback to understand why. Students mostly said there is no real pressure to prepare (no grading), and one could always hope that the other members have prepared. Thus, we decided to introduce the already described graded home assignments. The effect cannot be directly evaluated, but subjectively, students were prepared better for the hackathons.

### 5.2. Submission Time of the Hackathon

During the very first hackathons, we allowed the students to submit by the end of the day (i.e., by 23:59). However, supervision was provided only until 18:00. We wanted to give the students more time to finalize their projects and show them a realistic hackathon environment, where late-at-night coding is part of the culture. However, some students complained that this would keep them up at the university for too long; some groups also left the university and continued somewhere else. Some groups also tried to cheat and get help from outside. Thus, we decided to limit the non-supervised time to only 1 h and set the deadline to 19:00. Again, subjectively, this time works better.

### 5.3. Group Sizes for the Hackathon

During the first hackathons, we decided on groups of 4–5 students each. However, this number was logistically not optimal. Students tended to organize the work so that one is coding and the others are discussing and watching. Logistically, this does not work well for a group of 4–5 students, and some of the students were left out or started doing something else because they could not properly see what the other group members were doing. Thus, we limited the group sizes to three, which allowed the one in the middle to code and the other two to see well and discuss actively.

### 5.4. Hackathon Group Composition

In the beginning, we allowed students to self-organize into groups, and they tended to group by gender and by nationality. After discussing this with some teaching experts, we decided to control the group's formation and diversify the groups. However, we then observed that some students become shy and silent, mainly girls and non-proficient

German speakers. Thus, we went back to allow the students to self-organize to feel well in their groups and contribute and participate fully.

### 5.5. Explanatory Videos for Sample Exercises

Many students gave us feedback that they would appreciate solutions for the self-learning exercises. However, we did not want to provide them the solutions immediately. We made the experience that students tend to check the solution before attempting to solve the assignment independently. Furthermore, when solutions differed from their solutions, they tended to get confused that theirs is "wrong", even if both solutions were correct. Thus, in the end, we decided to offer them videos, where we would show how we would solve this exercise and explain all tricks and alternatives on the way. Even if the students watch the video without making an effort, they learn more than just quickly checking a piece of code. Students appreciated the videos, as they could also see our solution and coding process, not only the final result.

### 5.6. Complexity of Hackathon Assignments

One lesson we learned is that we need to set the complexity of hackathon assignments low. The psychological stress of having only limited time for the assignment and being graded is already a challenge for first-year students. Thus, otherwise, rather simple assignments become impossible to manage. Over the semesters, we gradually adjusted all assignments so that the students would have enough time to solve them and document them well. Our goal was for them to show what they have learned so far and deepen and extend their knowledge, and not unnecessarily stress them with overly long and complex assignments.

### 5.7. Hackathon Assignments and Online Sources

One important experience for us was how to prevent students from using Internet resources in the wrong way. On one side, we wanted to teach them how to efficiently use online resources to obtain more information, new solution ideas, etc. However, some of them tend to copy code snippets without understanding them. An example was an early version of Hackathon 11 (see again Figure 2), where we asked them to implement a breadth-first search (BFS) for a given graph. We received many solutions copied from online resources. However, most of the solutions did not work, as the students did not understand the functions or libraries they copied and did not manage to adapt them to the given graph. Some copied solutions were even in the wrong programming language.

These observations taught us to avoid such problems by not mentioning well-known algorithms such as BFS. Instead, we started inventing similar algorithms with invented names, e.g., calling a random walk "the lost student" walk or changing the rules of a game slightly and giving it a new name. Students still tried to find the solutions online but did not succeed and focused on solving the problem independently. It has to be noted here that we do not consider the discussed problem as cheating. Rather, it is exactly the right strategy to find a solution, but students need to build first their own experience before they can apply it efficiently.

### 5.8. Dependency between Hackathons

Another important lesson we learned is not to build too many dependencies between individual hackathons. In the first round of the summer semester, we designed Hackathon 11 to build upon Hackathon 10 and asked the students to visualize the given graph first. The ones who did not perform well during Hackathon 10 had a double problem now: they could not get past the visualization problem and did not have time to even start the new path problem. Now, we offer the visualization code as a template. All students can start with the template and can show that they have mastered the new topic.

## 6. Evaluation

In this section, we present some statistical data about our blended learning approach.

### 6.1. Preliminaries and Data Set

We have data from a total of four years or four cohorts. Two of the cohorts (the last two) were taught using the blended learning concept, while the other two cohorts had the same course contents but were taught using the traditional methods of lectures and tutoring sessions. The exact number of students per cohort can be seen in Figures 3 and 4.

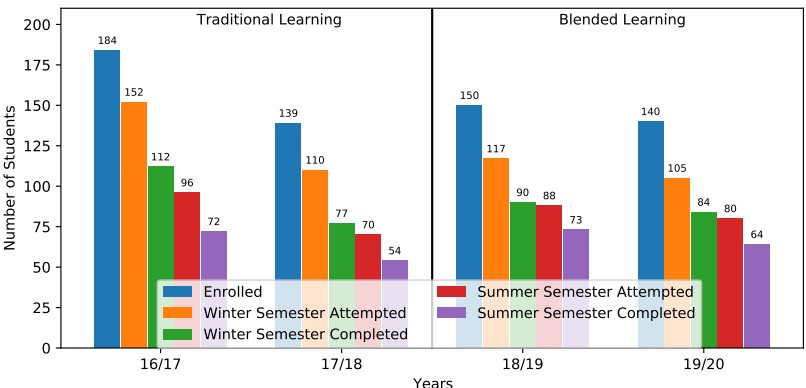

**Figure 3.** Course participation counts. The total number of students over the years remain more or less the same, except for the academic year 2016/2017.

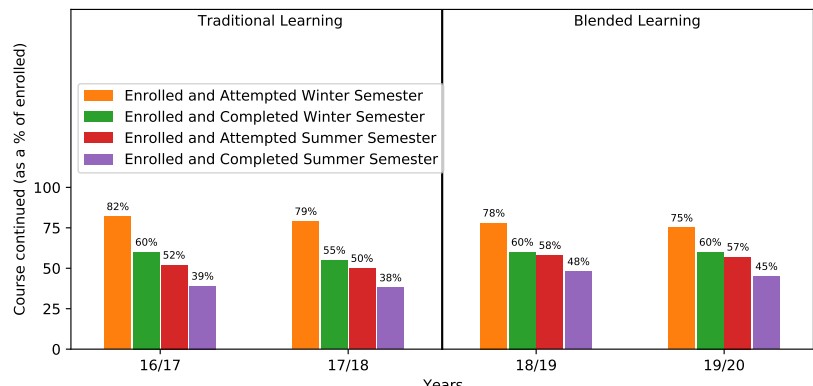

**Figure 4.** Course continuation percentages. With blended learning, less students abandon the course.

### 6.2. RQ1: Achieved Scores

The achieved grades or scores across the cohorts are presented in Figure 5. We can see the scores of the students achieved for homework assignments before the blended learning concepts and hackathons with the blended learning concept. Additionally, we can see the impact of bonus hackathons on the scores, e.g., for the year 2018/2019, the orange and red-spotted columns. We separately computed the scores with and without the bonus hackathons for a better understanding and evaluation of the data. The bonus hackathons count towards the final grades of the students but are not required. Instead, we used them to balance out sickness leaves or to give weaker students the chance to earn more points. Obviously, these improved the overall score of the students.

It can be seen that the scores did not change over the cohorts (if the bonus hackathons are not considered), with usual fluctuations. On the one side, we expected the scores to improve, which did not happen. On the other side, these results need to be evaluated in the context of more students remaining in the course. Thus, we not only managed to maintain the interest and motivation of students but also brought them to the same average score as their colleagues. We also evaluated the distributions of scores over the years, but no significant insights were observable.

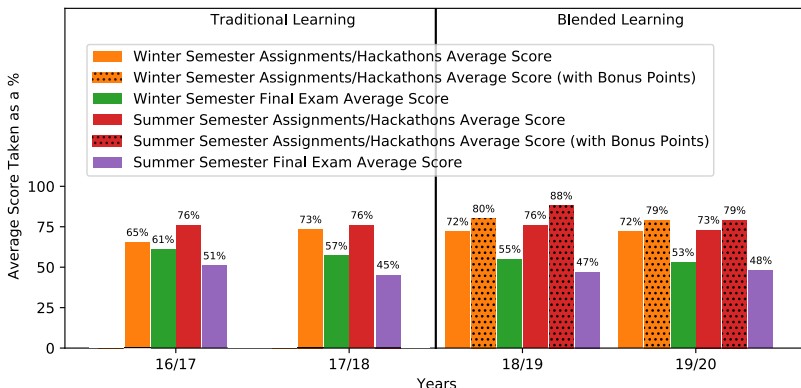

**Figure 5.** Average student scores over the different cohorts before and after implementing the blended learning concept. The spotted columns represent the hackathon scores with bonus hackathons considered, which improve the students' average scores.

### 6.3. RQ2: Fewer Students Drop the Course

The first important observation we made was that fewer students dropped the course after introducing the blended learning approach. This can be seen in Figures 3 and 4. The total number of students over the academic years does not significantly change and is due to a different number of students enrolled in EE in general.

With traditional teaching, only around 38–39% of the initially enrolled students finished the course. This number is low in itself, but it is considered normal in Germany. The German system allows to freely select courses and to plan individual schedules so that students can adapt better to other tasks, such as parenting, care or half-time jobs. Many students enroll in many courses and then select which ones to take after some weeks. Binding registrations are only due towards the end of the semester.

However, after we introduced the blended learning approach, approximately 45–48% of the students finished both semesters. Feedback from students shows that this is primarily due to higher motivation, more flexibility and less mandatory presence.

### 6.4. RQ3: More Bachelor Projects

Admittedly, this research question has little to do directly with the learning achievements of the students. However, it is a good indicator of their general interest in what we are teaching and how they perceive their future working opportunities in this area. Table 1 summarizes the number of bachelor projects throughout the years at our department. The numbers are low in general, but we see a clear trend in the last two years towards more projects. Consider that students do not start their bachelor theses simultaneously; thus, the numbers reported cannot be put directly into correlation with enrollment. However, looking at the enrollment numbers throughout the years in Figure 3, they even decrease slightly. Unfortunately, also no data are available on the total number of bachelor theses over the years. We also cannot evaluate the effect of the COVID-19 pandemic on these numbers.

**Table 1.** Number of bachelor projects at the Communication Networks Department by years of completion.

| 2016 | 2017 | 2018 | 2019 | 2020 | 2021 (Until 2006/2021) |
|------|------|------|------|------|------------------------|
| 3 | 1 | 3 | 3 | 6 | 7 |

We also evaluated the difference in continuation percentages and scores for different sub-groups, e.g., gender or study programs. However, we did not discover any significant differences either.

*6.5. Students' Feedback*

After each semester, we asked students for feedback via an approach called teaching analysis poll (TAP) [22]. In this method, students' discussions about the course are moderated by didactics experts who have not been part of the course. A few general questions are defined for the students beforehand to foster the discussions and to avoid concentrating only on problems. The students identify together positive or negative statements and refine them in small groups, with a focus on providing helpful feedback to be used for improvement, as opposed to "complaining". This method is beneficial, even if it does not provide quantifiable results such as grades or scores.

In our case, the three questions provided to the students in all semesters are listed below (translated from German). The questions are deliberately selected to be general and to focus on three areas: good things to keep, biggest problems, and concrete improvement ideas. The answers of all four semesters are quite long. Here, we offer some interesting insights and confirm our ideas which were new or unknown to us:

1.  What is most enhancing your learning in this course?
    - Self-paced learning.
    - I am not a native German speaker, and I can use subtitles on the videos and can repeat them as often as I want.
    - Good practical examples.

2.  What is most hindering your learning in this course?
    - Usage of difficult words in the videos.
    - Enforcing mixed teams.
    - Examples in the videos are rather simple, while exercises very complex.
    - Not enough concrete feedback about exercises and what was wrong.
    - In the hackathons, only one can really program/type; the others are watching.
    - No videos about debugging, problems and mistakes.
    - Searching for information on your own does not work.

3.  What suggestions do you have for improving this course?
    - Provide links where information can be found (instead of self-search).
    - Sample solutions for complex exercises.
    - Program more games.
    - Provide feedback faster.
    - More videos than text.
    - More text than videos.

The last two points (which provide more videos than text and vice versa) are especially typical for the outcome of such evaluations. Everybody learns at a different pace and with different methods. We already addressed most of the issues mentioned herein and discussed them in Section 5. In the next section, we discussed some of the remaining ones and how we plan to address them.

## 7. Discussion and Next Steps

The above results show that we achieved good results for RQ2 and RQ3, while the results for RQ1 are inconclusive. However, as discussed already above, we need to consider that fewer students now drop the course, which also changes the average scores. Additionally, while the first blended learning cohort in 2018/2019 achieved slightly better scores than the cohorts before, we observe a slight decrease again in 2019/2020. This might also be due to the COVID-19 pandemic, which influenced the second (summer) semester of the cohort with completely online work, including for the hackathons.

In general, we were delighted with the results of our blended learning use case. We have more and more intensive contact with the students and receive concrete and helpful feedback. Students can learn from course materials and directly from us by observing and coding with us during the hackathons. Last but not least, teaching the course has become

a more fulfilling job. However, there are still some components and details which can be optimized and adapted better to the course's learning goals.

### 7.1. Final Exam

As we noted before in Section 4, we kept the final exams after each semester as it is for comparison reasons. However, the results we obtained from two rounds of blended learning (see Section 6) and profound analysis of the exam questions lead us to the question of whether we test what we teach. The exam questions are targeted instead to a deep understanding of programming principles and fine details. However, our learning objectives mainly target not mastering a particular programming language or principle but rather "surviving" or "achieving results". We already started changing the exam questions by introducing practical coding questions with code submissions, however, the COVID-19 crisis prevented us from implementing it. Nonetheless, we will introduce the new questions in the next round of the course.

Additionally, the "I got it working" mentality cannot be our primary goal, and we would like to foster a profound understanding of programming principles in students. In [23], the authors have developed a "thinkathon" to address this problem. It consists of sets of pen-and-paper exercises to master programming principles and fine details and be worked on together with tutors during rather long sessions. We plan to incorporate this idea in the next round of our course.

### 7.2. Hackathon Assignments

For simplicity, we mostly repeated the hackathon assignments during the two rounds of our course, except for some adaptations as described earlier. However, this leads to some problems, i.e., students getting the solutions from the year before. Thus, in the following issues of the course, we plan to regularly change the hackathon assignments. Additionally, we plan to publish earlier assignments with sample solutions online for students to learn.

### 7.3. Introducing the Growth Mindset

The growth mindset is a concept from psychology, which postulates that people can profit and immensely grow from their mistakes and experiences as long as they take them as learning opportunities. This thinking is opposed to the fixed mindset when people believe their talents are fixed, and consequently, a failure in a test or any other sub-optimal result means they do not have "talent" for this topic. Obviously, the fixed mindset is not helpful when learning something new, and we observed this attitude very often in our classroom. When learning comes easily, students are motivated and excited. However, when it is difficult, as it is for some students without any programming background, they declare it is not their topic and even start complaining why this course is part of their EE studies at all.

The growth mindset has already been successfully applied in other CS courses, primarily by changing the feedback we give to students and putting more emphasis on their learning process and less on the grades and results they achieve [24]. We plan to apply this relatively simple but hopefully effective technique by following students' learning process over the modules closely and giving them personalized feedback about their learning progress. Currently, the challenge is purely technical and organizational: the system we use for evaluating and grading the students (StudIP, https://www.studip.de, accessed on 15 July 2021) is organized per assignment and not per student, and we thus do not have a good overview of the achievements and the development of individual students.

## 8. Conclusions

This paper has presented our experience with designing, building, and running a large two-semester long course with blended learning. In general, we are very satisfied with the results. One of the main challenges we faced was applying inverted classroom teaching without the risk of students not following the content at all, which is a typical experience

with pure inverted classroom approaches. We addressed this successfully by mixing the inverted classroom with the graded hackathons, which keeps the pace of the course. We believe this is the main and most important lesson for us: apply a mixed approach, well aligned with the individual challenges and requirements of the course.

Our experiences and results are also very valuable for other teachers. We show that blended learning is a very promising teaching concept in computer science and engineering, able to reduce drop-out rates, increase interest and motivation in students, and achieve even better scores. However, it also requires careful planning and continuous development via students' feedback—as any other teaching method, which focuses on the students and their learning process.

All our videos from the course are available under YouTube (https://www.youtube.com/ComNetsBremen, accessed on 15 July 2021). A complete copy of the winter semester 2019/2020 course is publicly available as a gitbook (https://comnets.gitbook.io/grundlagen-der-informatik-1-arduino-und-c/, accessed on 15 July 2021) in German.

We plan to re-design all our courses similarly: Defining first the learning goals, identifying challenges and restrictions, and then designing a mixed concept. We believe this general approach can also be applied in other disciplines and courses.

**Author Contributions:** Conceptualization, A.F. (Alexander Förster), L.K., J.M., L.G. and F.R.; methodology, A.F. (Alexander Förster), L.K. and J.M.; software, A.U., J.M., L.G. and J.K.; formal analysis, A.U.; resources, J.D., D.H. and L.G.; data curation, A.F. (Anna Förster), J.D. and A.U.; writing—original draft preparation, A.F. (Anna Förster) and F.R.; writing—review and editing, J.D., A.U., A.F. (Alexander Förster), D.H. and F.R.; supervision, A.F. (Anna Förster); project administration, A.F. (Anna Förster) and F.R.; funding acquisition, A.F. (Anna Förster) and F.R. All authors have read and agreed to the published version of the manuscript.

**Funding:** This research was funded partially by the ForstA-digital project at the University of Bremen, 2018–2020.

**Conflicts of Interest:** The authors declare no conflict of interest.

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
