# Peer review of "A Blended Learning Approach for an Introductory Computer Science Course"

_education, doi:10.3390/educsci11080372_

Round 1

Reviewer 1 Report

This is an interesting and meaningful analysis of research issues, but still, I have some doubts to be explained. 

1. I suggest the author better explain the conclusion of this study, such as the significance of this study for practice, as well as the significance and limitations of this study. 
2. Regarding previous work, I think there is a lot of recent important work left out and many references included that could be left out.
3. Did you had a proof reading by a native speaker.

Author Response

Dear reviewer,

Thank you for your time and efforts. Your comments are highly constructive and helpful and we believe they improved the quality of our work. We hope we have been able to address all of them and are looking forward to your further comments and remarks. Our answers are marked with a >>> for better visibility. In the manuscript, we have marked all new passages in RED.

Kind regards,

The authors.

--

This is an interesting and meaningful analysis of research issues, but still, I have some doubts to be explained. 

  1. I suggest the author better explain the conclusion of this study, such as the significance of this study for practice, as well as the significance and limitations of this study. 

>>> We have largely re-structured our manuscript, following the comments of all three reviewers. We have identified more clearly our research questions and provide extensive discussions on our results and conclusions in Sections 6, 7, and 8.

  1. Regarding previous work, I think there is a lot of recent important work left out and many references included that could be left out.

>>> This is right and has been improved in this new version. We re-structured the paper to include a separate section on related works and background (Section 2) and have included many new references.

  1. Did you had a proof reading by a native speaker.

>>> Yes, we have used a spell-checked and a native speaker to double-check our manuscript.

Reviewer 2 Report

This paper looks at a course design for early CS concepts to EE students.

Major comments:

  • The introduction needs to be an introduction.  The abstract is treated as the introduction and section 1 jumps right into the work.  This is a little jarring for the average academic reader and the structure needs to be revised
  • Some terms are neither defined or cited.  For example, Blended learning - I'm not sure I know what the definition is or who came up with the idea first.  I am familiar with inverted, but I think that not every reader will.
  • For the background, the paper needs to cite what has been done in this space before.  For example, who has used Python or Arduino in education - lots of people.  Who else has done 1st year CS for EEs?  What were their approaches?  What is blended and who has used it?
  • I think the paper should be structured a little different.  I would do Learning Outcomes, Constraints (Challenges), Course Description.  Also, the assessment process needs more details and how a hackathon differs compare to labs or assignment needs to be detailed more.
  • Results don't seem that convincing of why this approach is better.  It seems like the statistics are slightly better.
  • For a more major contribution to the field of education, I believe that materials should be made open-source or available to other instructors.  The paper is a "we did it, here it is, here is how it worked".  This adds little to the community of educators, and since the results show that "it might be better", then for the contribution to be useful, then at least, there should be a "and here are our materials so you can try this without having to build it from scratch".   Otherwise, the authors need to convince me of why this helps our educational community.
  • Learning outcomes should have some attachment to Bloom Taxonomy and a comment should be made that most of them are at a very low level, which is "understand"
  • Why are all the assignments and exercises not shown in relation to Figure 1 and Figure 2.  How do I understand the structure of the 2 activities without knowing how they are distributed.  Assessment is a major aspect of education, and so I need to understand how we provide feedback to the learner.
  • Growth mindset seems to be included without much lead.  Is it appropriate in this work?  If it is, why is it not in the background and explained in the design of the course?
  • Was the same multiple-choice exam used over the 4 years and it seems like a poor way of evaluating design concepts.  This should be commented on.

Minor comments:

  • I would change "freshmen" to "1st-year".  It has a more global understanding and has less from the USA's antiquated terminology.
  • A few minor english corrections.  For example, "the solutions at once" has no meaning.  "Statistical data" - is not what you mean - I think.  "very intuitive" ... the very is meaningless.  "covers a lot of topics" suggests that there is a list I can see.  
  • CS concepts - binary and hexadecimal are mathematical bases of counting.  They are represented in a machine which is a computing concept.
  • It seems like API and OO techniques is mixed up.  There are APIs that are OO based, and using them follows a specific syntax.

Author Response

Dear reviewer,

Thank you for your time and efforts. Your comments are highly constructive and helpful and we believe they improved the quality of our work. We hope we have been able to address all of them and are looking forward to your further comments and remarks. Our answers are marked with a >>> for better visibility. In the manuscript, we have marked all new passages in RED.

Kind regards,

The authors.

--

This paper looks at a course design for early CS concepts to EE students.

Major comments:

  • The introduction needs to be an introduction.  The abstract is treated as the introduction and section 1 jumps right into the work.  This is a little jarring for the average academic reader and the structure needs to be revised

>>> We have revised the complete structure of the article, including a proper introduction. The structure has been revised too.

  • Some terms are neither defined or cited.  For example, Blended learning - I'm not sure I know what the definition is or who came up with the idea first.  I am familiar with inverted, but I think that not every reader will.

>>> New references have been included, covering also blended learning and inverted classroom. Those terms are now also explained in Section 1.

  • For the background, the paper needs to cite what has been done in this space before.  For example, who has used Python or Arduino in education - lots of people.  Who else has done 1st year CS for EEs?  What were their approaches?  What is blended and who has used it?

>>> A new section 2 now includes a proper related works discussion.

  • I think the paper should be structured a little different.  I would do Learning Outcomes, Constraints (Challenges), Course Description.  Also, the assessment process needs more details and how a hackathon differs compare to labs or assignment needs to be detailed more.

>>> We have re-structured the paper according to the comments of all three reviewers.

  • Results don't seem that convincing of why this approach is better.  It seems like the statistics are slightly better.

>>> We have more clearly defined our research questions now in Section 3 and we have discussed our results in Section 6 in respect to the research questions.  It is right that the results for one of the research questions (Increasing students’ scores) are not conclusive, but the other two research questions showed good results. Furthermore, we have added some of the students’ feedback in Section 6.5, which also shows the clear preference of students to the new method.

  • For a more major contribution to the field of education, I believe that materials should be made open-source or available to other instructors.  The paper is a "we did it, here it is, here is how it worked".  This adds little to the community of educators, and since the results show that "it might be better", then for the contribution to be useful, then at least, there should be a "and here are our materials so you can try this without having to build it from scratch".   Otherwise, the authors need to convince me of why this helps our educational community.

>>> We agree that the availability of materials would be very valuable. Our videos are all available under YouTube and a copy of the complete part 1 of the course (the Arduino part) has been released as a gitbook. The links are provided in the conclusion. The second part (Python) will follow closely, but we need to manually copy-paste from our university system to something available publicly. Please note that the materials are in German, but we still believe that they could be of great use to the community.

  • Learning outcomes should have some attachment to Bloom Taxonomy and a comment should be made that most of them are at a very low level, which is "understand"

>>> This is a very good idea and we added a paragraph in Section 3.2 and we make use of this categorization in Section 3, which is also completely new.

  • Why are all the assignments and exercises not shown in relation to Figure 1 and Figure 2.  How do I understand the structure of the 2 activities without knowing how they are distributed.  Assessment is a major aspect of education, and so I need to understand how we provide feedback to the learner.

>>> We are not completely sure what you mean here. The grading of the course is detailed in Section 4. The structure of the individual blocks is also explained in detail there. Feedback is provided through the university e-learning system, StudIP.

  • Growth mindset seems to be included without much lead.  Is it appropriate in this work?  If it is, why is it not in the background and explained in the design of the course?

>>> Growth mindset is also mentioned as a future idea in Section 7.3. This is the reason why we have not further detailed it out. The reason for this is simple and unfortunate: our e-learning system does not provide us with a proper overview of the progress of your students – the feedback and scores are organized by assignment and not by student. We have now explained this problem in Section 7.3 and we hope to be able to address it asap, as we agree that it would tremendously improve the learning experience of our students.

  • Was the same multiple-choice exam used over the 4 years and it seems like a poor way of evaluating design concepts.  This should be commented on.

>>> Yes, it was the same, this is explained also in Sections 4 and 7.1

Minor comments:

  • I would change "freshmen" to "1st-year".  It has a more global understanding and has less from the USA's antiquated terminology.

>>> corrected.

  • A few minor english corrections.  For example, "the solutions at once" has no meaning.  "Statistical data" - is not what you mean - I think.  "very intuitive" ... the very is meaningless.  "covers a lot of topics" suggests that there is a list I can see.  

>>> corrected.

  • CS concepts - binary and hexadecimal are mathematical bases of counting.  They are represented in a machine which is a computing concept.

>>> As we explain in Section 3.2, these concepts need to be part of our course.

  • It seems like API and OO techniques is mixed up.  There are APIs that are OO based, and using them follows a specific syntax.

>>> Probably you mean the following sentences in Section 3.2: " Python enables us to teach the students basics of object oriented programming. Furthermore, it is basis for many popular libraries, such as numpy, matplotlib, TensorFlow, and many others. 
”. The two sentences are complementary to each other – we teach them OO and how to use libraries with APIs.

Reviewer 3 Report

Dear authors, thank you for your submission. Here are some considerations that I think could improve your article.

The article deals with a case of redesigning a subject from a more traditional system to a mixed system in which the Flipped Classroom and Blended Learning approach are combined.

The subject matter is of great interest. It is an innovation in a course that takes into account the profile of the students, their diversity and their needs and motivations and, finally, it results in a reduction of the dropout rate. This is all the more interesting in today's pandemic context.

The writing of the article is very agile, easy to understand; the article is well structured and the content is of great interest. However, there is a great imbalance between the theoretical part and the empirical part. I would like to point out some issues that I consider essential to introduce and others that could be improved:

  • I would remove the "C" from Keywords, as it is not significant.
  • The research question of the article is not made explicit in the introduction.
  • An essential section is missing, that of the methodology used. It could be a case study. In this case, its use should be justified and its limitations pointed out.
  • The Filpped Classroom and Blended Learning methodologies are the protagonists of innovation in the subject. However, no theoretical framework is provided, nor are their advantages and limitations pointed out in relation to the case studied. A synthesis of the literature in relation to the two methodologies would be needed.
  • The conclusions should be written in such a way as to answer the research question.
  • The abstract should contain the research question and the method used, as well as the results and discussion.
  • References 3 and 4 are not well formulated.
  • The references are somewhat sparse. They should be completed with those of the suggested theoretical synthesis. It is important that they should be of recent date.

In summary, the article is of great interest at the present time but should be methodologically strengthened.

Author Response

Dear reviewer,

Thank you for your time and efforts. Your comments are highly constructive and helpful and we believe they improved the quality of our work. We hope we have been able to address all of them and are looking forward to your further comments and remarks. Our answers are marked with a >>> for better visibility. In the manuscript, we have marked all new passages in RED.

Kind regards,

The authors.

--

Dear authors, thank you for your submission. Here are some considerations that I think could improve your article.

The article deals with a case of redesigning a subject from a more traditional system to a mixed system in which the Flipped Classroom and Blended Learning approach are combined.

The subject matter is of great interest. It is an innovation in a course that takes into account the profile of the students, their diversity and their needs and motivations and, finally, it results in a reduction of the dropout rate. This is all the more interesting in today's pandemic context.

>>> Thank you.

The writing of the article is very agile, easy to understand; the article is well structured and the content is of great interest. However, there is a great imbalance between the theoretical part and the empirical part. I would like to point out some issues that I consider essential to introduce and others that could be improved:

  • I would remove the "C" from Keywords, as it is not significant.

>>> Done.

  • The research question of the article is not made explicit in the introduction.

>>> We have now clearly defined your research questions in Section 3.3.

  • An essential section is missing, that of the methodology used. It could be a case study. In this case, its use should be justified and its limitations pointed out.

>>> The methodology has been defined now also in the new Section 3.3.

  • The Filpped Classroom and Blended Learning methodologies are the protagonists of innovation in the subject. However, no theoretical framework is provided, nor are their advantages and limitations pointed out in relation to the case studied. A synthesis of the literature in relation to the two methodologies would be needed.

>>> We have provided more details about the two new concepts and relevant literature in the Introduction (completely re-written) and in a new Section 2.

  • The conclusions should be written in such a way as to answer the research question.

>>> We provided a discussion based on the research questions in Section 6.

  • The abstract should contain the research question and the method used, as well as the results and discussion.

>>> We believe the abstract should not be too long – instead, we extended the results section and provided new sections 2 and 3. A short summary of the methods and results is already provided in the abstract.

  • References 3 and 4 are not well formulated.

>>> corrected.

  • The references are somewhat sparse. They should be completed with those of the suggested theoretical synthesis. It is important that they should be of recent date.

>>> We have included a new Section 2 with new references.

In summary, the article is of great interest at the present time but should be methodologically strengthened.

Round 2

Reviewer 1 Report

Congratulations, I accept this publication

Author Response

Thank you for your time and efforts!

Reviewer 2 Report

The revisions have made the paper significantly better.  I still have a few comments below for what I think is one more revision cycle to get the work to a point that I believe it is ready for the next steps.

Comments:

  • "well how lectures are"?
  • "At the same time,"?  Is time related here?
  • The english is still quite rough - two examples above.  I would run the paper through "word" and "grammarly" to remove many of the simple problems.
  • Table 1 could give a % as well of possible to show that there's just not an increase in the pool.
  • Citations missing
    • 2011 - "Arduino for Teaching Embedded Systems. Are Computer Scientists and Engineering Educators Missing the Boat?"
    • 2011 - "Python for Teaching Introductory Programming: A Quantitative Evaluation"

Response - response:

  • CS concepts - binary and hexadecimal are mathematical bases of counting. They are represented in a machine which is a computing concept.

>>> As we explain in Section 3.2, these concepts need to be part of our course.

I don't disagree, but they are part of computer science as they are other fields...  The concepts come from mathematics and are used in a range of fields.

Author Response

Dear reviewer, please see below our answers to your comments. Thank you very much again for your time and effort.

  • "well how lectures are"?
  • "At the same time,"?  Is time related here?
  • The english is still quite rough - two examples above.  I would run the paper through "word" and "grammarly" to remove many of the simple problems.

>>> We have followed your suggestion and checked the complete text with advanced tools. 

  • Table 1 could give a % as well of possible to show that there's just not an increase in the pool.

>>> We understand the problem and the issue raised. However, there is no solid possibility to put these numbers in correlation with each other - we have added the following explanation in the manuscript: "Consider that students do not start their bachelor theses simultaneously; thus, the numbers reported cannot be put directly into correlation with enrollment. However, looking at the enrolment numbers throughout the years in Figure 3, they even decrease slightly. Unfortunately, also no data is available on the total number of bachelor theses over the years. "

  • Citations missing
    • 2011 - "Arduino for Teaching Embedded Systems. Are Computer Scientists and Engineering Educators Missing the Boat?"
    • 2011 - "Python for Teaching Introductory Programming: A Quantitative Evaluation"

>>> Thank you for the suggestions, we have added them. 

Response - response:

  • CS concepts - binary and hexadecimal are mathematical bases of counting. They are represented in a machine which is a computing concept.

>>> As we explain in Section 3.2, these concepts need to be part of our course.

I don't disagree, but they are part of computer science as they are other fields...  The concepts come from mathematics and are used in a range of fields.

Reviewer 3 Report

English should be revised. 

Author Response

We have revisited the complete manuscript. Thank you very much again for your time and effort.